

# Relative telomere length and oxidative DNA damage in hypertrophic ligamentum flavum of lumbar spinal stenosis

Sinsuda Dechsupa[1], Wicharn Yingsakmongkol[2], Worawat Limthongkul[2], Weerasak Singhatanadgige[2] and Sittisak Honsawek[1]

[1] Osteoarthritis and Musculoskeleton Research Unit, Department of Biochemistry, Faculty of Medicine, King Chulalongkorn Memorial Hospital, Thai Red Cross Society, Chulalongkorn University, Bangkok, Thailand
[2] Department of Orthopaedics, Faculty of Medicine, King Chulalongkorn Memorial Hospital, Thai Red Cross Society, Chulalongkorn University, Bangkok, Thailand

## ABSTRACT

**Background**. Lumbar spinal stenosis (LSS) is a common cause of low back pain with degenerative spinal change in older adults. Telomeres are repetitive nucleoprotein DNA sequences of TTAGGG at the ends of chromosomes. Oxidative stress originates from an imbalance in pro-oxidant and antioxidant homeostasis that results in the production of reactive oxygen species (ROS). The purpose of this study was to investigate relative telomere length (RTL) and oxidative DNA damage in ligamentum flavum (LF) tissue from LSS patients.

**Methods**. Forty-eight patients with LSS participated in this study. Genomic DNA from non-hypertrophic and hypertrophic LF tissue were analyzed by real-time polymerase chain reaction for relative telomere length (RTL). 8-hydroxy 2′-deoxygaunosine (8-OHdG) levels were determined by using enzyme-linked immunosorbent assay. We cultivated LF fibroblast cells from patients in different ages (61, 66, and 77 years). After each cultivation cycle, we examined RTL and senescence-associated β-galactosidase (SA-β-gal) expression.

**Results**. The hypertrophic LF had significantly lower RTL than non-hypertrophic LF ($P = 0.04$). The levels of 8-OHdG were significantly higher in hypertrophic LF compared to non-hypertrophic LF ($P = 0.02$). With advancing cell culture passage, the number of cells in each passage was significantly lower in hypertrophic LF fibroblast cells than non-hypertrophic LF fibroblast cells. When evaluated with SA-β-gal staining, all senescent LF fibroblast cells were observed at earlier passages in hypertrophic LF compared with non-hypertrophic LF fibroblast cells.

**Discussion**. Our results showed that patients with LSS displayed an accelerated RTL shortening and high oxidative stress in hypertrophic LF. These findings implied that telomere shortening and oxidative stress may play roles in the pathogenesis of hypertrophic LF in lumbar spinal stenosis.

Corresponding author
Sittisak Honsawek,
sittisak.h@chula.ac.th

## INTRODUCTION

Lumbar spinal stenosis (LSS) is one of the most common spinal pathologies in the elderly and results in buttock or lower extremity pain, radiculopathy, neurogenic claudication, and cauda equina syndrome. LSS has been one of the leading indications for lumbar spinal surgery in patients older than 60 years of age. The finding of LSS is characterized by a decrease of elastin-to-collagen ratio, age-related fibrosis, and hypertrophic ligamentum flavum (*Katz & Harris, 2008*). Ligamentum flavum (LF) is a yellow ligament containing high collagen and elastin fibers. Elastin provides the elasticity, while collagen fiber provides the tensile strength and stability of the spine (*Losiniecki et al., 2013*; *Li, Xia & Han, 2015*; *Yabe et al., 2015*). With age and degeneration, an increase in collagen and a decreased elastin-to-collagen ratio were evident in hypertrophied LF (*Mobbs & Dvorak, 2007*); however, the underlying mechanisms remain unknown. Environmental, biochemical, and genetic risk factors may be involved in the pathogenesis of LSS. A number of cytokines have been postulated to play major parts in the fibrosis of LF in LSS. In recent years, the increased expressions of various matrix metalloproteases, connective tissue growth factor, basic fibroblast growth factor, and inflammatory cytokines have been highlighted in the hypertrophy of LF (*Park et al., 2009*; *Zhong et al., 2011*; *Honsawek et al., 2013*; *Park et al., 2013*). Telomere length and oxidative stress including DNA damage have never been investigated in lumbar spinal stenosis.

Telomeres are repetitive nucleoprotein DNA sequences of TTAGGG at the ends of eukaryotic chromosomes (*Diotti & Loayza, 2011*; *McEachern, Krauskopf & Blackburn, 2000*). They protect chromosomal ends from DNA double-strand breaks and prevent them from DNA degradation (*Houben et al., 2008*; *Lingner & Cech, 1998*). Telomeres in human somatic cells decline with each round of cell proliferation both in vitro and in vivo due to the inabilities of conventional polymerases to fully replicate the parenting DNA by lagging strand synthesis (the end replication problem) (*O'Sullivan & Karlseder, 2010*). Eventually, this can lead to telomere shortening and cell cycle arrest at the G1 phase, with consequences being senescent cells and apoptosis (*Zhu, Belcher & Van der Harst, 2011*). One senescence marker is senescence-associated β-galactosidase (SA-β-gal) staining, which increases with cellular aging (*Kim et al., 2008*; *Kim et al., 2009*; *Takahashi et al., 2004*). Several studies have shown telomere shortening in various conditions including degenerated intervertebral discs, keloid, and biliary atresia (*Le Maitre, Freemont & Hoyland, 2007*; *De Felice, Wilson & Nacca, 2009*; *Udomsinprasert et al., 2015*).

Oxidative stress results from an imbalance in pro-oxidant and antioxidant homeostasis that leads to the production of high levels of reactive oxygen species (ROS) and/or reactive nitrosative species (RNS) (*Lu, 2007*). Moreover, exogenous factors can produce ROS and/or RNS, including UV radiation, smoking, and carcinogenic substances (*Dröge, 2002*). High levels of ROS have been demonstrated as oxidative damage to lipids of cell membranes, proteins, and DNA damage to human cells (*Valavanidis, Vlachogianni & Fiotakis, 2009*). Oxidative DNA damage has been produced by hydroxyl radical (OH$^\bullet$) from various mechanisms and lead to the interaction of OH$^\bullet$ with genomic base, which is forming 8-hydroxy 2′-deoxyguanosine (8-OHdG) (*Valko et al., 2004*; *Kroese & Scheffer,*

*2014*). 8-OHdG can be a biomarker for oxidative stress in carcinogenesis, aging, and cell senescence and can be found in human samples, such as blood leukocytes, tissues, and urine (*Sajous, Botta & Sari-Minodier, 2008*). LSS may be induced by oxidative stress, leading to DNA damage and nucleobase instability. Until now, the possible roles of telomere length and oxidative DNA damage, including the SA-β-gal for cell senescence, have never been investigated in LSS patients.

Telomere erosion and increasing oxidative stress may be involved in degenerative conditions and cellular aging. We hypothesized that telomere shortening and oxidative DNA damage could be associated with the hypertrophic LF in LSS patients. Accordingly, the objectives of this study were to investigate relative telomere length (RTL) and oxidative DNA damage in LF tissue from LSS patients. We further cultivated human LF fibroblast cells from patients of different age for in vitro cell culture to determine RTL and evaluate senescence markers as SA-β-gal expression.

## MATERIALS AND METHODS

### Study population

LF specimens were obtained from 48 lumbar spinal stenosis patients (19 male and 29 female) who underwent decompression laminectomy for neurogenic claudication. The mean age of patients was $63.3 \pm 9.7$ years (range, 45.0–87.0 years). The hypertrophic LF specimens were obtained from hypertrophic LF levels in each patient which mostly occurred at L3/L4 and L4/L5. The non-hypertrophic LF was collected from the same patients who served as a control group. All LF specimens were pooled in 1.5 mL microcentrifuge tubes containing 200 μL of RNAlater (Qiagen, Hilden, Germany) and kept at $-80$ °C until measurement. All human LF specimens were obtained with the written informed consent of the patients who participated in this study. All of the consent procedures and experimental protocols were approved by the Institutional Review Board on Human Research of the Faculty of Medicine, Chulalongkorn University (IRB 259/59).

### Isolation and culture of human LF cells

Hypertrophic and non-hypertrophic LF specimens were classified into three age groups as follows: 61, 66, and 77 years. LF cells were isolated as described in a previous study (*Jeong, Lee & Kim, 2014*). Whole LF tissue was minced into pieces of approximately 0.2 cm into T25 flasks. Cells were maintained in complete medium (Alpha-Modified Eagle's medium with 10% fetal bovine serum, 1% penicillin-streptomycin (GIBCO—BRL, Grand Island, NY, USA)) at 37 °C in humidified atmosphere containing 5% $CO_2$ until they grew to confluence. Cells were fed by complete replacement of medium every three days. When the primary cultures reached confluence, adherent cells were detached using 0.25% trypsin EDTA solution (GIBCO–BRL) and seeded at a density of $1.0 \times 10^5$ cells in T25 flasks containing complete medium. When reaching confluence, cells were trypsinized and passaged sequentially in the same manner until 10 passages. Throughout the experiment, human LF cells were cultivated under the standard culture condition.

## Determination of relative telomere length

Genomic DNA was extracted directly from LF tissues and cells according to the instruction of the DNA extraction kit (Vivantis, Buckinghamshire, Malaysia). The concentration of DNA samples was measured using the Nanodrop 2000 spectrophotometer (Thermo Scientific, Wilmington, DE, USA). The relative telomere length was determined by quantitative real-time PCR method, as previously described by *Cawthon (2002)*. Telomere length was measured according to the ratio of the telomere repeat copy number (T) to the single-copy gene copy number (S) in each given sample. The single-copy gene refers to the 36B4 gene, which encodes the acid ribosomal phosphoprotein (PO). The ratio (T/S) is proportional to the average telomere length. To begin with, the RTL was analyzed by using SYBRGreen Master Mix none-ROX (RBC Bioscience, Taipei, Taiwan). The primers for the repeat copy number were 10 μM telomere forward (5′-CGGTTTGTTTGGGTTTGGGTTTGGGTTTG GGTTTGGGTT-3′) and 10 μM telomere reverse (5′-GGCTTGCCTTACCCTTACCCTTACCCTTACCC TTACCCT-3′) and for single-copy gene (36B4) PCR were 10 μM 36B4-forward (5′-CAGCAAGTGGGAAGGTGTAATCC-3′) and 10 μM 36B4-reverse (5′- CCCATTCTATCA TCAACGGGTACAA-3′). The profile of the telomere and single copy gene amplification started with 95 °C incubation for 10 min followed by 40 cycles of 15 s at 95 °C and 1 min at 54 °C.

All amplification was performed on a StepOnePlus Real Time PCR system (Applied Biosystems, Foster City, CA, USA). Each PCR sample was run in duplicate using 3.1 ng DNA per 10 μL reaction. The targeted gene (Tel, ND1) PCR reaction was performed on spate runs with the same samples, and melting curve result was expressed for every run to verify specificity.

## In vitro replicative lifespan and growth rate of LF cells measured by cumulative population doubling level (PDL) and population doubling rate (PDR)

At each subcultivation, confluent hypertrophic and non-hypertrophic LF cells were trysinized, counted, and reseeded at a density of $1.0 \times 10^5$ cells in T25 flask. An increase in the cumulative population doubling at each subcultivation was calculated as follows: population doubling (PD) = $[\log_{10}(N_H) - \log_{10}(N_I)]/\log_{10}(2)$, where $N_H$ = the number of harvested cells and $N_I$ = the number of seeded cells (*Cristofalo et al., 1998*). The calculated population doubling increase was then included in the previous population doubling level (PDL), to yield the cumulative population doubling level. Moreover, to determine growth rate, the PDRs of hypertrophic and non-hypertrophic LF cells cultured in vitro were measured as follows: PDR = the PDL/the number of days in culture (*Stenderup et al., 2003*).

## Senescence associated-β-galactosidase staining (SA-β-gal)

At each subcultivation, hypertrophic and non-hypertrophic LF cells were taken for SA-β-gal staining. The percentage of SA-β-gal-positive LF cells was determined by the previous method (*Kim et al., 2009*). Briefly, LF cells were seeded in six-well plates at a cell density of $5.0 \times 10^4$ cells per well containing complete medium and incubated at 37 °C for 24 h. Cells were washed in phosphate buffer saline (PBS) and fixed with 2% formaldehyde and

0.2% glutaraldehyde in 10X PBS for 15 min at room temperature. Then, cells were rinsed with PBS and SA-β-gal staining solution (Cell Signaling Technology, Danvers, MA, USA) was added that contained 40 mM citric acid/sodium phosphate (pH 6.0), 150 mM NaCl, 2 mM mgCl$_2$, 5 mM potassium ferrocyanide, 5 mM potassium ferricyanide, and 1 mg/mL of 5-bromo-4-chloro-3-indoyl-β-D-galactopyranoside at 37 °C for 16 h. After incubation, 300 LF cells in random fields were counted and the percentage of SA-$β$-gal-positive LF cells was calculated by using the following formula: (the number of SA-β-gal-positive LF cells/300 LF cells) × 100. Cell count was repeated twice and the mean was calculated.

### Determination of 8-OHdG concentration

Total DNA was extracted directly from LF tissue. Firstly, approximately 200 mg of LF tissue was homogenized with liquid nitrogen and added with 10 µl proteinase K and 400 µl lysis buffer containing 50 mM tris-hydrochloride (pH 7.4), 1 mM ethylenediaminetetraacetic (pH 8.0), 0.5% w/v sodium dodecylsulfate and incubated at 50 °C for 2 h. After incubation, homogenous LF tissue was added with 250 µL phenol and 250 µL chloroform:indole-3-acetic acid (CHCl$_3$:IAA; 49:1). The lysate was centrifuged at 13,500 rpm for 30 min at 4 °C. The supernatant was carefully transferred into a tube containing 4 µL glycogen, 40 µL sodium acetate, 800 µL absolute ethanol and kept overnight at −20 °C. The lysate was centrifuged at 13,500 rpm for 30 min at 4 °C. The supernatant was washed with 1 mL 70% ethanol and centrifuged at 13,500 rpm for 5 min at 4 °C. The supernatant was transferred into a microcentrifuge tube and put into a vacuum machine for 15 min and dissolved with distilled water. Total DNA concentration was determined using a NanoDrop® ND-100 Spectrophotometer (Thermo Fisher Scientific, Hampton, NH, USA) and 200 µg/mL of adjusted DNA in each sample. The 8-OHdG concentration of LF tissue was performed by quantitation of 8-OHdG in LF tissue DNA using HT 8-oxo-dG enzyme-linked immunosorbent assay kit (Trivigen, Gaithersburg, MD, USA) according to the manufacturer's instructions.

### Statistical analysis

All statistical analyses were performed with the SPSS statistical package, version 20.0 (SPSS Inc., Chicago, IL, USA). All values were reported as mean ± standard error of the mean (SEM). Paired $t$-test was used to compare the means of two dependent groups. Comparison between the two groups was employed by the Wilcoxon signed-rank test when the distributions were not among the groups. A $P$-value less than 0.05 (based on a two-tailed test) was considered statistically significant.

## RESULTS

### RTL and 8-hydroxy-2′-deoxyguanosine levels of LF tissue samples in LSS patients

We investigated RTL in 48 non-hypertrophic and 48 hypertrophic LF tissues from LSS patients. As shown in Fig. 1, RTL in LF tissue samples were significantly lower in hypertrophic LF tissue than in non-hypertrophic LF tissue (1.01 ± 0.07 vs. 1.15 ± 0.08; $P = 0.04$). To investigate status of oxidative damage in LSS patients, we determined

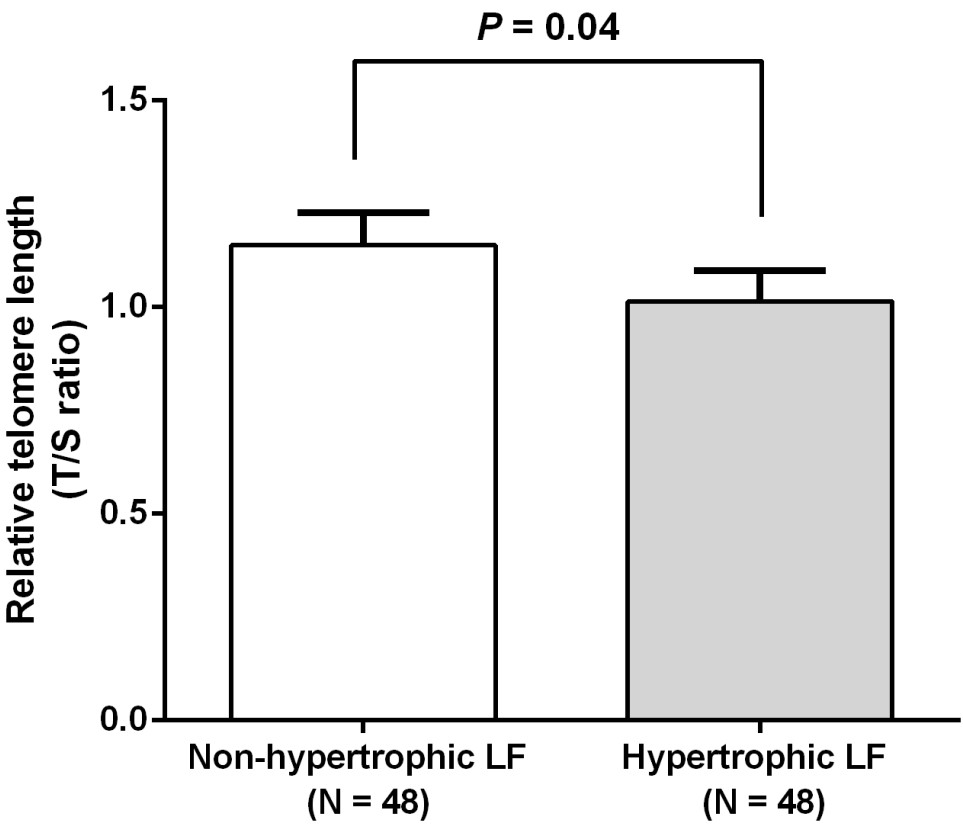

*P* = 0.04

**Figure 1** Comparison of relative telomere length of non-pathologic and pathologic ligamentum flavum tissues from LSS patients.

8-OHdG levels in 30 non-hypertrophic and 30 hypertrophic LF tissue samples in LSS patients. The mean 8-OHdG levels of LF tissue were significantly higher in hypertrophic LF tissue than in non-hypertrophic LF tissue ($0.43 \pm 0.03$ vs. $0.64 \pm 0.09$; $P = 0.02$), as demonstrated in Fig. 2.

### RTL in LF fibroblast cells

Real-time PCR analysis was used to determine the RTL of the non-hypertrophic and hypertrophic LF fibroblast cells obtained from patients 61, 66, and 77 years old. There was a significant decline in telomere length with advancing passages (Fig. 3). RTL of the hypertrophic LF fibroblast cells was significantly lower than that of the non-hypertrophic LF fibroblast cells in the first passage ($4.17 \pm 0.43$ vs. $5.85 \pm 0.19$; $P = 0.04$).

### Cumulative PDL and PDR

The mean PDL of hypertrophic and non-hypertrophic LF cells obtained from patients aged 66 years were $4.84 \pm 0.40$ and $4.44 \pm 0.51$, respectively (Fig. S1). The mean PDR of hypertrophic and non-hypertrophic LF cells were $0.14 \pm 0.03$ and $0.12 \pm 0.02$ PD/day, respectively (Fig. S2).

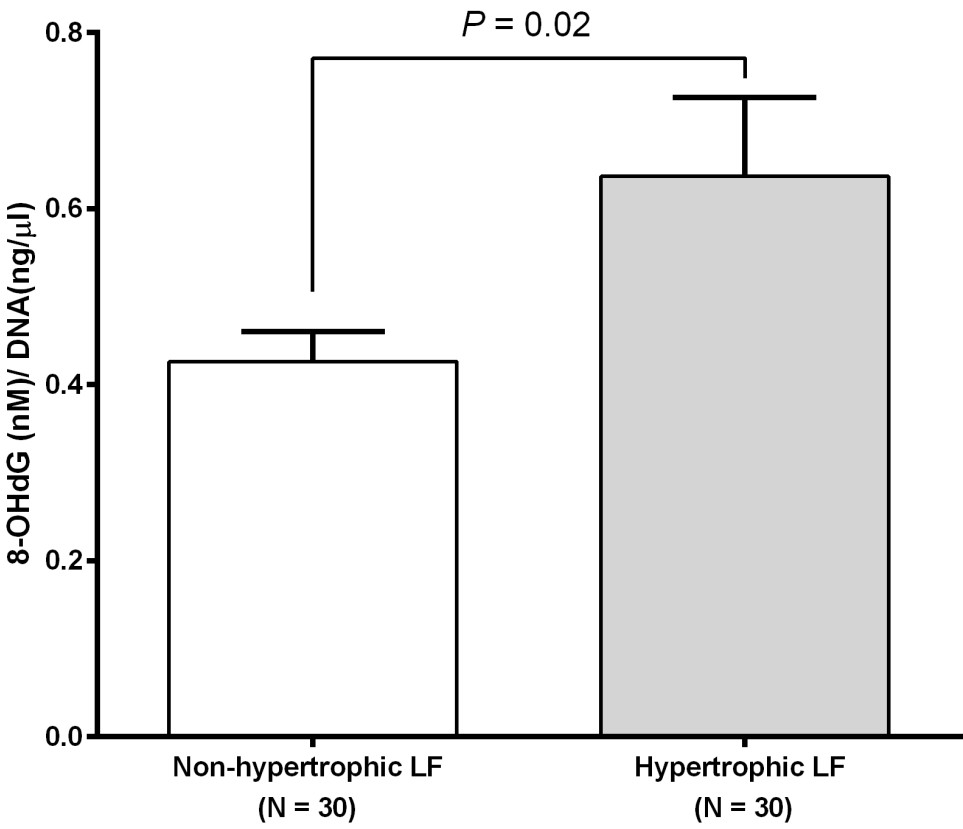

**Figure 2** Ligamentum flavum tissue 8-hydroxy-2′-deoxyguanosine concentration of non-pathologic and pathologic ligamentum flavum tissues from LSS patients.

### SA-β-gal staining

The mean of percentages of non-hypertrophic and hypertrophic SA-β-gal-positive LF fibroblast cells from patients aged 61 and 66 years at passage 1 were 0.30%, 0.00%, 0.30%, and 0.30% respectively, which steadily increased with advancing passages (Fig. 4). Meanwhile, the mean of percentages of non-hypertrophic and hypertrophic SA-$\beta$-gal-positive LF fibroblast cells from patients aged 77 years, even at passage 1, were 2.70%, and 6.30%, which reached more than 80% at passage 10. The SA-β-gal staining study showed that SA-β-gal-positive LF fibroblast cells was increased in hypertrophic LF compared to non-hypertrophic LF fibroblast cells (Fig. 5).

### DISCUSSION

Lumbar spinal stenosis is a common cause of low back pain caused by degenerative changes that are common in older adults. The pathophysiology consists in compression within the central canal and in changes in facet joints, disc bulging, and hypertrophic LF (*Azimi et al., 2015*). Previous studies have shown that the thickness of LF corresponds with the decrease of elastic and collagen fibers and the increase of stiffness and fibrosis (*Park et al., 2005*; *Chen et al., 2014a*; *Chen et al., 2014b*). miR-155 has been recently demonstrated to increase

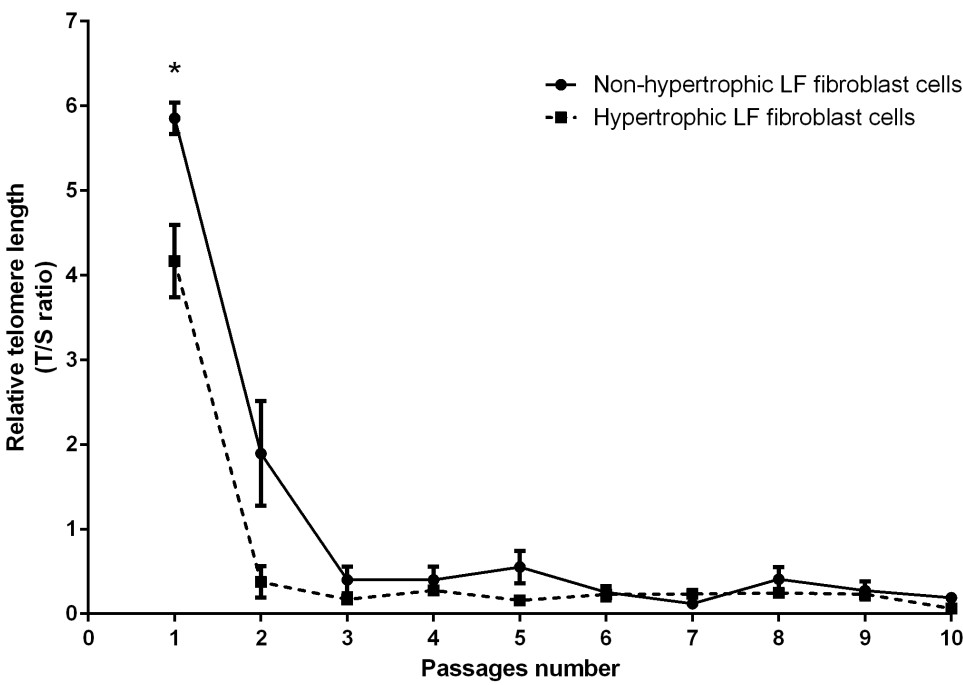

**Figure 3   Relative telomere length of non-pathologic and pathologic LF fibroblasts cells from LSS patients aged 61, 66, and 77 years.**

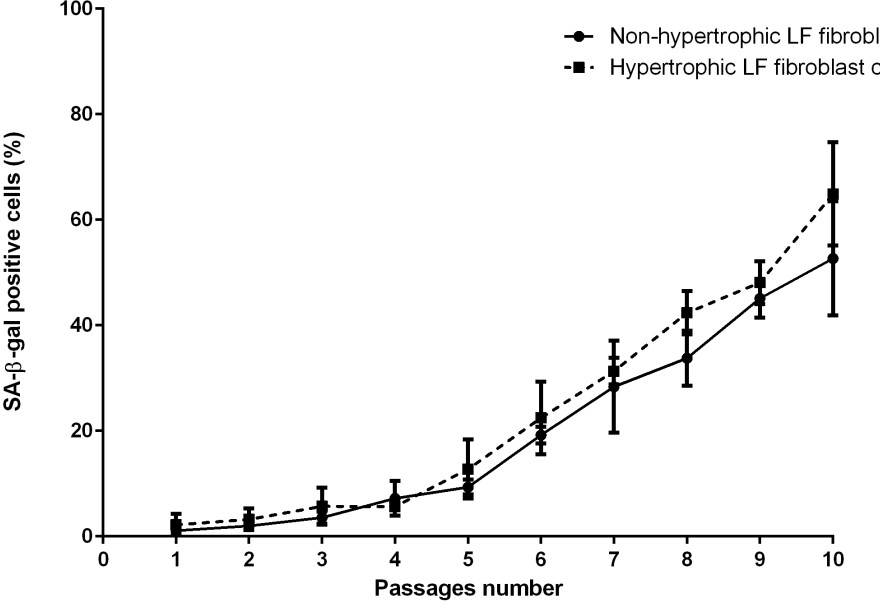

**Figure 4   The percentage of the senescence-associated-β-galactosidase (SA-β-gal) positive of non-pathologic and pathologic LF fibroblasts cells from LSS patients aged 61, 66, and 77 years.**

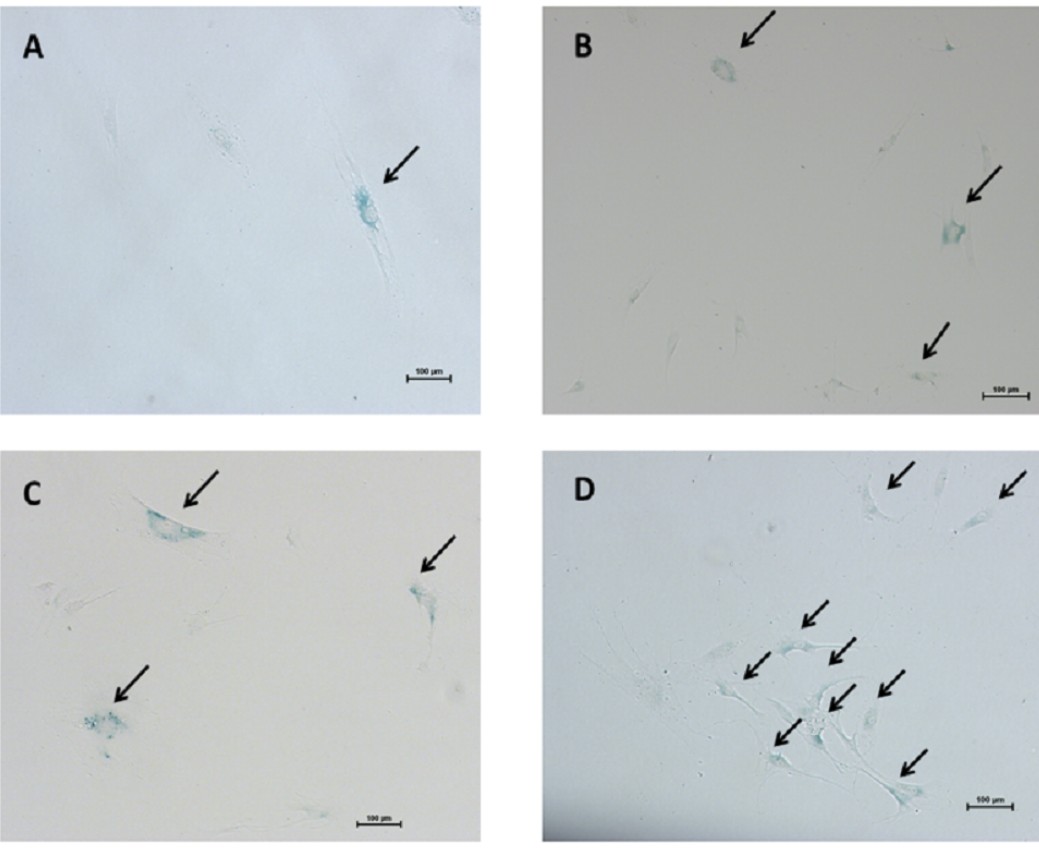

**Figure 5** **The senescence-associated-β-galactosidase (SA-β-gal) positive pathologic LF fibroblasts cells as shown in arrows.** Representative staining from patients aged 61 years (A) passage 1; (B) passage 10 and patients aged 77 years (C) passage 1; (D) passage 10.

the expression of type I and type III collagen in fibroblasts of LF and correlate with the degree of fibrosis of LF in LSS patients (*Chen et al., 2014a*; *Chen et al., 2014b*).

In the current study, we investigated RTL and the levels of oxidative DNA damage of non-hypertrophic and hypertrophic LF tissues and cell fibroblasts in LSS patients and evaluated senescent cells by SA-β-gal marker. To the best of our knowledge, this is the first study to show RTL in hypertrophic LF tissue in LSS patients. We showed that hypertrophic LF tissue from LSS patients presented shorter RTL than non-hypertrophic LF tissue, suggesting that the hypertrophic LF cells may exhibit high turnover rate of cell division (Fig. S1). This finding is accordant with the recent study, *Le Maitre, Freemont & Hoyland (2007)* found that the mean telomere length was significantly decreased in subjects with degenerative disc disease when compared with non-degenerative discs. We hypothesized that the rate of telomere shortening in LF fibroblasts would reflect the rate of cell turnover in the human LF tissues (Fig. S2).

In the present study, LF fibroblast cells exhibited different RTL compared to the parental LF tissues from which they have been isolated. The mechanism of shortened RTL in hypertrophic LF tissues is not easily addressed. A number of possible mechanisms either

independently or in combination, may be postulated. In LSS, hypertrophic LF tissues contain various kinds of cells including fibroblasts, inflammatory cells, and mesenchymal stem cells. Recently, LF precursor cells were isolated from the hypertrophic LF tissues and these cells displayed features of mesenchymal stem cells (*Chen et al., 2011*). All these cell types in mixed populations have different *telomere* lengths. Our study relied upon average *RTL* in *cell populations* or the entire tissue and the results would be masked when *mixed populations* in LF tissues are measured. Moreover, chronic repetitive inflammation and oxidative stress could contribute to telomere shortening in hypertrophic LF tissues. It should be noted that a variety of these factors may act in concert to generate the phenomenon. The mechanism behind the connection of different RTL in hypertrophic LF tissue and in hypertrophic LF fibroblast cells remains a mystery and requires further study.

Telomere erosion and DNA damage are the most important of aging-related factors. Environmental stimuli in degenerative diseases may also be causing cell senescence. In our study, senescence marker analysis revealed greater numbers of SA-$\beta$-gal positive cells in hypertrophic LF fibroblastic cells with advancing culture passages of elderly patients with LSS. These findings are also in line with a previous study by *Jeong, Lee & Kim (2014)*, where disc cells from older patients had a lower telomere length and reached to senescence in vitro earlier than those from younger patients with intervertebral disc disease. The results indicate that patient age is a major factor. Cells from elderly patients might have lower levels of telomerase activity and senescent cells accumulate with age.

An increased generation of ROS and radicals in cellular redox level has been associated with degenerative diseases and aging (*Venkatesh et al., 2012*). The 8-OHdG marker is measured as an indicator of oxidative DNA damage. The current study showed that hypertrophic LF 8-OHdG values were significantly higher in hypertrophic LF than in non-pathologic LF. This finding is consistent with a previous study (*Tsai et al., 2010*) showing that cultured orbital fibroblasts from patients with Graves' ophthalmopathy had higher levels of 8-OHdG compared with normal controls. In our recent study (*Udomsinprasert et al., 2016*), we revealed that 8-OHdG levels were significantly increased in the serum and liver specimens of biliary atresia patients compared with controls. The increase of oxidative stress condition may be causing DNA damage within cells involved in the pathogenesis of hypertrophic LF in LSS patients.

Several limitations need to be acknowledged. Firstly, we determined the relative telomere length and oxidative DNA damage in LSS patients, but did not measure the activity of telomerase enzymes. The mechanisms underlying altered telomerase expression in the hypertrophic LF remain unclear. Additional studies are warranted to further investigate the telomerase activity and SA-β-gal staining in LF fibroblast cells from the younger and older patients with LSS. Secondly, this study was cross-sectional in its design. Therefore, prospective longitudinal researches are necessary to demonstrate the association between RTL, 8-OHdG, and LF thickness. Moreover, we could not access normal LF tissues or LF fibroblasts from normal age-matched subjects without LSS, or any other normal comparable tissues, that would serve as a good control to determine RTL and 8-OHdG levels. Lastly, the sample size was relatively small, which made it challenging to analyze

data with different age decades of patients. Our power of statistical difference was relatively low. Thus, further studies with large numbers of patients are warranted.

## CONCLUSIONS

The present study demonstrated that patients with LSS had a shorter RTL and higher oxidative DNA damage in hypertrophic LF than in non-pathologic LF. The in vitro experiments showed that the number of senescence cells was increased in the hypertrophic LF fibroblasts. Although underlying mechanisms and pathways remain unclear and require additional studies, this study may shed some light on the pathogenesis of hypertrophic LF in lumbar spinal stenosis.

**Abbreviations**

| | |
|---|---|
| **LSS** | Lumbar spinal stenosis |
| **LF** | Ligamentum flavum |
| **8-OHdG** | 8-hydroxy-2′-deoxyguanosine |
| **SA-β-gal** | Senescence-associated β-galactosidase |

## ACKNOWLEDGEMENTS

The author would also like to thank the Research Core Facility of the Department of Biochemistry and Chulalongkorn Medical Research Center (ChulaMRC) for kindly providing facilities. The authors are gratefully thankful to Napaphat Jirathanathornnukul for technical assistance. We thank Professor Henry Wilde for reviewing and proof-reading the manuscript.

### Funding

This work was supported by the 90th Anniversary of Chulalongkorn University Scholarship and H.M. the King Bhumibhol Adulyadej's 72nd Birthday Anniversary Scholarship and the research chair grant from the National Science and Technology Development Agency (RES5829130016). The funders had no role in study design, data collection and analysis, decision to publish, or preparation of the manuscript.

### Grant Disclosures

The following grant information was disclosed by the authors:
90th Anniversary of Chulalongkorn University Scholarship.
King Bhumibhol Adulyadej's 72nd Birthday Anniversary Scholarship.
National Science and Technology Development Agency: RES5829130016.

### Competing Interests

The authors declare there are no competing interests.

## Author Contributions

- Sinsuda Dechsupa and Sittisak Honsawek conceived and designed the experiments, performed the experiments, analyzed the data, contributed reagents/materials/analysis tools, prepared figures and/or tables, authored or reviewed drafts of the paper, approved the final draft.
- Wicharn Yingsakmongkol, Worawat Limthongkul and Weerasak Singhatanadgige conceived and designed the experiments, performed the experiments, analyzed the data, authored or reviewed drafts of the paper, approved the final draft, sample collection.

## Human Ethics

The following information was supplied relating to ethical approvals (i.e., approving body and any reference numbers):

The Institutional Review Board on Human Research of the Faculty of Medicine, Chulalongkorn University granted Ethical approval to carry out the study (IRB 259/59).

## Data Availability

The raw data are provided in a Supplemental File.

## Supplemental Information

Supplemental information for this article can be found online at http://dx.doi.org/10.7717/peerj.5381#supplemental-information.

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
