# Peer review of "Relative telomere length and oxidative DNA damage in hypertrophic ligamentum flavum of lumbar spinal stenosis"

_PeerJ, doi:10.7717/peerj.5381_

## Round 0.1 · original submission · Major Revisions

The concerns of reviewer 2 in particular may require additional measurement or experimental work. Be sure to justify your response regarding these concerns.

Reviewer 1 ·

Basic reporting

No comments

Experimental design

No comments

Validity of the findings

No comments

Additional comments

Telomeres are very important for chromosome maintenance and contain highly conserved repetitive DNA sequence 5’TTAGGG 3’. It forms the ends of linear eukaryotic chromosomes and protects it from damage and all. The authors have studied the relative telomere length and oxidative DNA damage in ligamentum flavum tissue collected from 48 adult patients suffering from lumbar spinal stenosis. The authors have compared the relative telomere length from genomic DNA of non-hypertropic and hypertropic LF tissues using RT-PCR and also analyzed 8-OHdG levels, the marker for oxidative DNA damage. The authors also observed the positive senescence signal in pathologic LF fibroblast cells. The current study clearly demonstrates that the patients suffering from lumbar spinal stenosis has shorter RTL and higher oxidative DNA damage in hypertropic ligamentum flavum. This is a very novel finding and manuscript is suitable for PeerJ with minor changes.


Minor comments:

• Please reframe this statement: “Telomeres are nucleoprotein TTAGGG DNA sequences at the end of chromosomes.”

• Please add these reference regarding telomeres:
1) Lingner, J. and Cech, T.R. (1998) Telomerase and chromosome end maintenance. Curr. Opin. Genet. Dev. 8, 226–232.
2) McEachern, M.J., Krauskopf, A. and Blackburn, E.H. (200

• Please add the abbreviation of Reactive Oxygen Species (ROS) in the background section.

• Line # 70: Please elaborate the end replication problem of telomere.

• Line # 126: The concentration of DNA samples was measured…

• Line 129: Cawthon et al: reference is missing

• Please cite: Roderick J. O’Sullivan and Jan Karlseder, Nat Rev Mol Cel Biol. (2010). 11(3): 171-181: “Reference: Hypertrophy of Ligamentum Flavum in Lumbar Spine Stenosis Is Associated with Increased miR-155 Level.”

• The authors should also include the increased expression and activities of various matrix metalloproteases, connective tissue growth factors, inflammatory cytokines etc. with proper references.

Reviewer 2 ·

Basic reporting

Minor suggestions:
1. Briefly explain what LSS is, in the introduction.
2. Define T/S ratio briefly in the materials and methods section.
3. Mention lysis buffer constituents in the materials and methods section.
4. Include the citation for Cawthon et al. (for the technique followed to determine RTL) in the references section.

Experimental design

In this manuscript titled “Relative telomere length and oxidative DNA damage in hypertrophic ligamentum flavum of lumbar spinal stenosis” the authors examine the role of telomere shortening and oxidative stress as possible mechanisms underlying lumbar spinal stenosis pathogenesis. The authors present data to claim that hypertrophic ligamentum flavum (LF) obtained from patients with lumbar spinal stenosis (LSS) has lower relative telomere length (RTL) and higher oxidative stress compared to non-hypertrophic LF samples obtained from the same patients, using whole DNA extracts. Furthermore, the authors present data showing lower RTL and higher senescence associated β-galactosidase in fibroblasts isolated from hypertrophic LF tissues of specific patients.

This manuscript provides some evidence of accelerated senescence and oxidative stress induced DNA damage in hypertrophic LF tissues of patients with LSS compared to normal LF tissues of the same patients.

Some noteworthy aspects of the manuscript:

1. The authors have provided detailed descriptions of the experimental methods and provided sufficient background information relating to the study.
2. Use of non-hypertrophic LF tissues from the same patients as negative controls.
3. Acknowledging the limitations of the study in an honest manner.

Validity of the findings

1. From Figure 1, the RTL of non-hypertrophic LF tissues is marginally higher than hypertrophic LF tissues and from Figure 2, the 8-OHdG levels is marginally lower in non-hypertrophic LF tissues than the hypertrophic counterparts. This phenomenon is constant in all age group of LSS patients used in the study (i.e. from age 45-87- data on supplemental S1-sheet 1).

Although use of non-hypertrophic LF tissue samples from the same patient is a good negative control to show that telomere shortening and oxidative damage is accelerated during hypertrophic conditions, it is possible that these non-hypertrophic tissues (especially if obtained from the lumbar area) are showing early signs of telomere shortening/oxidative damage that can result in low statistical significance. This can be seen from the fact that the RTL of non-hypertrophic and hypertrophic LF fibroblasts is comparable after passage 3 (Figure 3).

If the authors have access to normal LF tissues or LF fibroblasts from normal age-matched people with no LSS, or any other normal comparable tissues, that would serve as a good control to determine normal RTL and 8-OHdG levels.

2. The authors have used fibroblasts from patients aged 61, 66, and 77 years to examine RTL changes during passaging. The RTL represented by T/S ratio in the first passage of hypertrophic cells from the above patients are 3.36, 4.32, and 4.82 respectively (Supplemental S1- sheet2). However, the RTL (T/S ratio) of the hypertrophic LF tissue from the same patients is 1.61, 1.43, and 1.19 respectively (Supplemental S1-sheet1). Can the authors explain why LF cells in culture have different RTL compared to the parent tissues from which they have been isolated?

3. In the discussion, paragraph 2, the authors state “We showed that hypertrophic LF tissue from LSS patients presented shorter RTL than non-hypertrophic LF tissue, suggesting that the hypertrophic LF tissue may exhibit high turnover rate of cell division.”

The authors are encouraged to present data comparing the growth rate of hypertrophic and non-hypertrophic LF fibroblasts.

4. In the Discussion, paragraph 3, the authors state “Cells from elderly patients might have lower levels of telomerase activity and senescent cells accumulate with age.” In the absence of data about telomerase levels, can the authors include β-galactosidase data for the patients in the 45-55 age group? Are they different from the older patients i.e. patients >60 years?

---

## Round 0.2 · Minor Revisions

As you certainly understand, I sent your first revision to one of the original reviewers. I think that you should be able to submit a new revision very quickly as it requires only some wording changes. I look forward to receiving your revised manuscript soon.

Reviewer 2 ·

Basic reporting

1. I recommend the authors to refer to the Supplementary data on the growth rates in the Results, and Discussion sections (e.g. line 227, 232) and also include the experimental method in the Materials and Methods section.

2. In the discussion, line 228, the statement ".... suggesting that the hypertrophic LF tissue may exhibit high turnover rate of cell division" needs to be revised to change the word "tissue" to "cells" to reflect the supplementary growth data.

Experimental design

The authors have included additional data on the growth rates to add validity to the statement that hypertrophic LF cells exhibits higher rate of cell division compared to non-hypertrophic LF cells.

Validity of the findings

No comments

Additional comments

The authors have sufficiently addressed the reviewers' concerns. The manuscript is suitable for publishing. However, I recommend the authors to refer to the Supplementary data on the growth rates in the Results, and Discussion sections and include the experimental method in the Materials and Methods section.

---

## Round 0.3 · accepted · Accept

Please do note that the reviewer suggested referring to the Supplementary Information at appropriae places in the Results section. Thank you for the revised manuscript.